

# Sex differences in gene expression and alternative splicing in the Chinese horseshoe bat

Wenli Chen, Weiwei Zhou, Qianqian Li and Xiuguang Mao

School of Ecological and Environmental Sciences, East China Normal University, Shanghai, China

## ABSTRACT

Sexually dimorphic traits are common in sexually reproducing organisms and can be encoded by differential gene regulation between males and females. Although alternative splicing is common mechanism in generating transcriptional diversity, its role in generating sex differences relative to differential gene expression is less clear. Here, we investigate the relative roles of differential gene expression and alternative splicing between male and female the horseshoe bat species, *Rhinolophus sinicus*. Horseshoe bats are an excellent model to study acoustic differences between sexes. Using RNA-seq analyses of two somatic tissues (brain and liver) from males and females of the same population, we identified 3,471 and 2,208 differentially expressed genes between the sexes (DEGs) in the brain and liver, respectively. DEGs were enriched with functional categories associated with physiological difference of the sexes (e.g.,gamete generation and energy production for reproduction in females). In addition, we also detected many differentially spliced genes between the sexes (DSGs, 2,231 and 1,027 in the brain and liver, respectively) which were mainly involved in regulation of RNA splicing and mRNA metabolic process. Interestingly, we found a significant enrichment of DEGs on the X chromosome, but not for DSGs. As for the extent of overlap between the two sets of genes, more than expected overlap of DEGs and DSGs was observed in the brain but not in the liver. This suggests that more complex tissues, such as the brain, may require the intricate and simultaneous interplay of both differential gene expression and splicing of genes to govern sex-specific functions. Overall, our results support that variation in gene expression and alternative splicing are important and complementary mechanisms governing sex differences.

## INTRODUCTION

Sex differences in phenotypes (*e.g.*, morphology, physiology and behavior) are quite common across a wide range of sexually reproducing organisms. Most of sexually dimorphic traits can be achieved by differential gene expression between the sexes, defined as sex-biased gene expression (*Ellegren & Parsch, 2007*). In the last two decades, sex-biased gene expression has been extensively studied in numerous species including humans, and these studies have shown that sex-biased gene expression is present ubiquitously among

Corresponding author
Xiuguang Mao,
xgmao@sklec.ecnu.edu.cn

different tissues in these organisms (*Rinn & Snyder, 2005*; *Ingleby, Flis & Morrow, 2015*; *Mank, 2017*), including human (*Mayne et al., 2016*; *Oliva et al., 2020*).

Alternative splicing (AS), as another important form of gene regulation, is a widespread phenomenon among eukaryotes (*Kim, Magen & Ast, 2007*) and contributes greatly to the complexity of organisms and adaptive evolution by creating multiple proteins from a single gene (*Nilsen & Graveley, 2010*; *Singh & Ahi, 2022*). Because males and females largely share an identical genome, sex-biased AS can act as an alternative mechanism, relative to sex-biased gene expression, to produce sexually dimorphic traits, in particular when pleiotropic constraints limit changes of gene expression level (*Rogers, Palmer & Wright, 2021*). Indeed, sex-specific AS has been documented in a number of animal species, *e.g.*, *Drosophila* (*Telonis-Scott et al., 2009*; *Gibilisco et al., 2016*), primate (*Blekhman et al., 2010*), fish (*Naftaly, Pau & White, 2021*), and human (*Karlebach et al., 2020*). However, very few studies have attempted to investigate the relative roles of differential gene expression and alternative splicing in sexual differences of animals (but see *Rogers, Palmer & Wright, 2021*; *Singh & Agrawal, 2021*).

Bats belong to the order Chiroptera and comprise over 1400 species (*Simmons & Cirranello, 2020*). Similar to other mammals, bats also exhibit many sexually dimorphic traits (*Camargo & De Oliveira, 2012*; *Grilliot, Burnett & Mendonça, 2014*; *Stevens & Platt, 2015*; *Wu et al., 2018*). Most of the studies, in bats, focused on sex differences in echolocation pulse frequency (reviewed in *Siemers et al., 2005*) due to its important role in communication of bats (*Jones & Siemers, 2011*). Horseshoe bats are one of the most popular groups to study acoustic differences between sexes because they emit constant frequency (CF) in echolocation calls which can be assessed accurately by researchers (*Siemers et al., 2005*).

In this study, using one horseshoe bat (*Rhinolophus sinicus*) as the system, we are the first to explore sex differences of gene regulation (differential gene expression and alternative splicing) in bats. Unlike most horseshoe bats showing overlap of call frequencies between sexes, *R. sinicus* exhibits non-overlap of sex differences (*Xie et al., 2017*; *Mao et al., 2013*). In addition, a high-quality chromosome-level genome has been generated for *R. sinicus* (*Ren et al., 2020*). This genomic resource can help to quantify transcript expression accurately and make it possible to perform alternative splicing analysis based on short-read RNA-seq data. Specifically, we collected bat individuals in April when they arouse from hibernation and start to feed extensively. For female bats, they also begin to prepare for reproduction. We propose that if the sex differences are largely encoded by sex-biased gene expression and/or alternative splicing, we expect to observe multiple differentially expressed or spliced genes between the sexes which are associated with acoustic difference, feeding or female reproduction. To test for our proposal, we obtained mRNA-seq data of brain and liver from four individuals of each sex. Brain is responsible for regulation of almost all life activities and was recently used to study acoustic differences between the sexes of frog (*Chen et al., 2022*). Liver is the primary organ for metabolism and is related to feeding. In addition, these two tissues have been commonly used to explore sex differences of gene expression and/or alternative splicing in other animals (*Naurin et al., 2011*; *Trabzuni et*

**Table 1  Detailed information of samples used in this study (modified from *Chen et al., 2022*).**

| Sample ID | Sex | Tissues | Sampling locality | Sampling date |
|---|---|---|---|---|
| 180404 | Male | Brain and liver | Jiangsu, China | April 19, 2018 |
| 180406 | Male | Brain and liver | Jiangsu, China | April 19, 2018 |
| 180411 | Male | Brain and liver | Jiangsu, China | April 19, 2018 |
| 180401 | Male | Brain and liver | Jiangsu, China | April 19, 2018 |
| 180402 | Female | Brain and liver | Jiangsu, China | April 19, 2018 |
| 180403 | Female | Brain and liver | Jiangsu, China | April 19, 2018 |
| 180409 | Female | Brain and liver | Jiangsu, China | April 19, 2018 |
| 180410 | Female | Brain and liver | Jiangsu, China | April 19, 2018 |

*al., 2013*; *Blekhman et al., 2010*; *Zheng et al., 2013*; reviewed in *Rinn & Snyder, 2005*). Thus, results from our current study may shed some light on sex-biased gene regulation in bats.

## MATERIALS & METHODS

### Sampling and mRNA-seq data collection

All samples used in this study were obtained from *Chen & Mao (2022)* and raw sequencing reads were available from the NCBI Sequence Read Archive (SRA) under Bioproject accession number PRJNA763734. Briefly, bats were captured using mist nets in Jiangsu, China in April (Fig. 1A and Table 1) and only adult bats were sampled. Bats were euthanized by cervical dislocation and tissues of brain and liver were collected for each bat. We chose four males and four females in transcriptomics analysis (Fig. 1B). All 16 tissues were frozen immediately in liquid nitrogen and stored in a −80 °C freezer. Sequencing libraries from 16 tissues were created with NEBNext® UltraTM RNA Library Prep Kit for Illumina® (NEB, USA) and sequenced on an Illumina HiSeq X Ten platform (paired-end 150 bp). Because *R. sinicus* is not in the list of state-protected and region-protected wildlife species in the People's Republic of China, no permission is required. Our sampling and tissue collection procedures were approved by the National Animal Research Authority, East China Normal University (approval ID Rh20200801).

### RNA-Seq data trimming and mapping

Following *Chen & Mao (2022)*, raw sequencing reads from each sample were processed using TRIMMOMATIC version 0.38 (*Bolger, Lohse & Usadel, 2014*) with the parameters of SLIDINGWINDOW:4:20. We further trimmed reads to 120 bp and removed those with <120 bp in order to meet the requirement of rMATs (see below) that all input reads should be of equal length. Then, filtered reads were mapped to a male *R. sinicus* reference genome (a chromosome-level genome with scaffold N50 of >100 Mbp and annotation of >20,000 genes, *Ren et al., 2020*) using HISAT2 version 2.2.0 (*Kim, Langmead & Salzberg, 2015*) with default settings. The resulting SAM files were converted to sorted BAM files with SAMtools v1.11 (*Li et al., 2009*). The mRNA alignments in sorted BAM files were used in both differential expression (DE) and alternative splicing (AS) analysis.

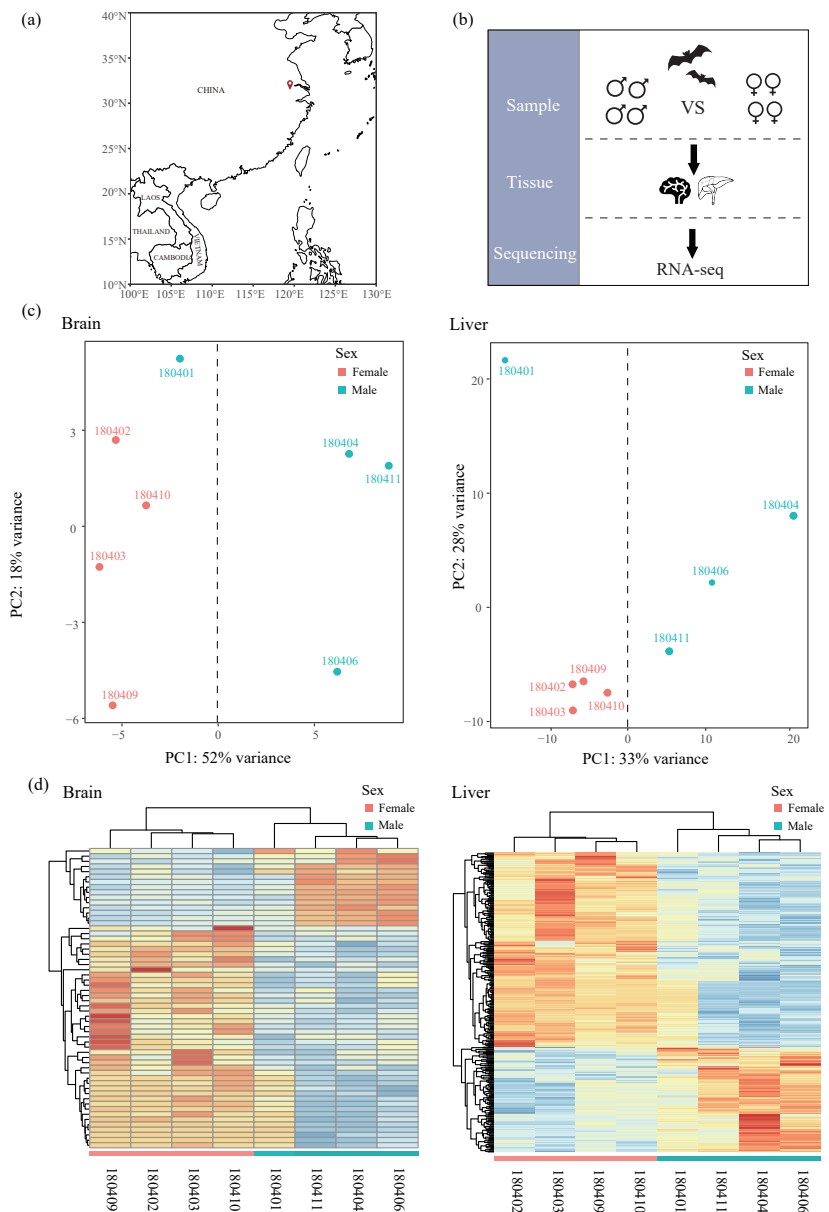

**Figure 1** **Sampling, experimental design and clustering analysis.** (A) Sampling locality in this study. (B) Experimental design. Bats of females and males were collected and compared based on RNA-seq data of two tissues (liver and brain). (C) Principal component analysis (PCA) based on normalized count matrix of all genes in the brain and liver. (D) Hierarchical clustering and heatmap based on normalized count matrix of all genes in the brain and liver.

## Differential expression analysis

Mapped reads in each sample were quantified using featureCounts (*Liao, Smyth & Shi, 2014*) with default settings and normalized across samples using DESeq2 (*Love, Huber & Anders, 2014*). To assess the similarity of expression patterns across samples in each tissue, we conducted a principal component analysis (PCA) using PlotPCA function in

DESeq2 package (*Love, Huber & Anders, 2014*). In addition, we also performed hierarchical clustering and heatmaps with the R package pvclust v2.2-0 (*Suzuki & Shimodaira, 2006*) and pheatmap v1.0.12 (*Kolde, 2012*), respectively. These two analyses on all samples of each tissue revealed one outlier (180401, Figs. 1C and 1D) which was excluded from the downstream analyses. For each tissue, we filtered out the lowly expressed genes with an average CPM (counts per million) < 1 among individuals of each sex. Then we identified sex-specific genes, including male-specific genes and female-specific genes, by comparing the list of genes expressed in each sex. After that, shared genes in both sexes were used to perform DE analysis with DESeq2 (*Love, Huber & Anders, 2014*) to identify sex-biased genes (SBGs), including male-biased genes (MBGs) and female-biased genes (FBGs). We determined SBGs with the $P$ value < 0.05 after Benjamini and Hochberg adjustment for multiple tests ($p$adj < 0.05, *Benjamini & Hochberg, 1995*). To investigate the grouping of samples based on expression patterns across genes, we performed hierarchical clustering and heatmaps based on Euclidean distances of rlog-transformed read counts of each SBG using the R package pvclust v2.2-0 (*Suzuki & Shimodaira, 2006*) and pheatmap v1.0.12 (*Kolde, 2012*), respectively. The reliability of each node in clustering was determined using bootstrap resampling (1,000 replicates).

Here, differentially expressed genes (DEGs) between males and females included both sex-specific genes and sex-biased genes (DEGs-female: female-specific genes and female-biased genes; DEGs-male: male-specific genes and male-biased genes).

## Alternative splicing analysis

rMATs (v4.1.0) (*Shen et al., 2014*) was used to identify the AS events between the sexes in each tissue. Five different types of AS events were detected by rMATs including skipped exons (SE), mutually exclusive exons (MXE), alternative 5′ and 3′ splice site (A5′SS and A3′SS), and retained intron (RI). rMATs assesses each splicing event by the PSI value (percent spliced-in value) which indicates the proportion of an isoform in one group to the other group at each splice site. Following *Rogers, Palmer & Wright (2021)*, AS events were determined using 0 < PSI < 1 in at least half of the samples in each group to reduce the false positives. To compare AS between groups, the inclusion difference (ΔPSI, average PSI of one group minus average PSI of another group) was calculated for each AS event. Following *Grantham & Brisson (2018)*, significance of ΔPSI between the two groups was determined using the false discovery rate (FDR) <0.05 and ΔPSI > 0.1. Genes with significant ΔPSI were considered as differentially spliced genes (DSGs).

To characterize the transcriptional similarity of splicing across samples in each tissue, we also performed hierarchical clustering and heatmaps based on Euclidean distances of the PSI value of each DSG using the R package pvclust v2.2-0 and pheatmap v1.0.12. Following *Rogers, Palmer & Wright (2021)*, when a gene has multiple splice events the average PSI value is used. Bootstrap resampling procedure was used to assess the reliability of each node (1,000 replicates).

## Chromosomal distribution of DEGs and DSGs

We test whether DEGs and DSGs were significantly enriched in X chromosome relative to the autosomes. We compared the observed number of DEGs and DSGs to the corresponding

expected number. Non-random distribution was estimated using Fisher's exact test and significance was determined using a $P$-value <0.05.

## Overlapping between DEGs and DSGs

We test for the overlap between DEGs and DSGs following *Rogers, Palmer & Wright (2021)*. Specifically, we first calculated the expected number of genes that are both DEGs and DSGs as (total no. DEGs × total no. DSG)/total no. expressed genes. Next, the representation factor (RF) was calculated to compare the observed number of overlapped genes to the expected number. RF > 1 and RF < 1 indicate more overlap than expected and less overlap than expected, respectively. We used a hypergeometric test in R version 4.0.5 (*R Core Team, 2021*) to test for significance of comparisons between the observed and expected overlaps. Significance was determined with a $P$-value <0.05.

## Functional gene ontology analysis

Metascape (http://metascape.org) was used to perform functional enrichment analysis on genes identified in DE and AS analyses with the Custom Analysis module (*Zhou et al., 2019*). A total of 13,905 expressed genes identified in this study were used as the background list. Significantly enriched gene ontology (GO) terms and KEGG pathways were determined with corrected $p$-value using the Banjamini-Hochberg multiple test correction procedure and $q$-value < 0.05. Log ($q$-value) of −1.3 is equal to $q$-value of 0.05. Redundancy was removed using the REVIGO clustering algorithm (http://revigo.irb.hr/) with the default settings. We then used the R ggplot2 package to visualize the clustered GO terms.

## RESULTS

Here, we obtained 16 tissue samples of RNA-seq data from *Chen & Mao (2022)* with an average of 39,217,309 filtered pair reads per sample and an overall alignment rate of 98.11% to the reference genome (Table S1). Based on these data, we identified and characterized the differentially expressed genes and spliced genes between males and females. We also compared these two sets of genes by exploring their distribution patterns in the genome and the extent of their overlap to assess their relative roles in sex differences.

## Identification and characterization of sex-specific genes

In the brain, we identified 232 female-specific and 133 male-specific genes among 13,456 expressed genes (Fig. 2A and Table 2). In contrast, we found more number of sex-specific genes in the liver (458 and 230, female-specific and male-specific genes, respectively) among 11,502 expressed genes (Fig. 2B and Table 2). Detailed sex-specific genes have been described in Table S2.

To explore the functional categories of the sex-specific genes, we performed functional enrichment analysis. In the brain, male-specific genes were enriched into 21 significant GO terms and three KEGG pathways and most of them were related to digestion, fatty acid or lipid transport, and histidine catabolic process (Fig. 2C and Table S3). For female-specific genes, although not significant after accounting for multiple testing ($q$-value > 0.05), they were enriched into several interesting terms with uncorrected $P$ < 0.01, such as nuclear

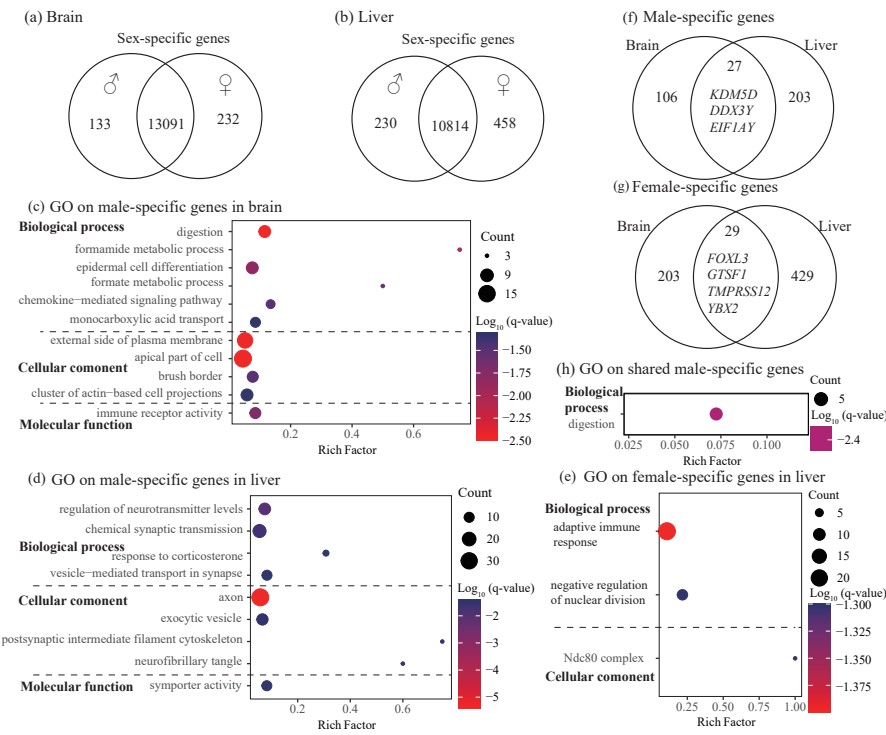

**Figure 2** **Identification and characterization of sex-specific genes.** (A–B) Venn diagrams showing sex-specific genes. (C–E) Significant Gene Ontology (GO) terms enriched on the sex-specific genes in the brain (C) and liver (D and E). (F–G) Venn diagrams showing the number of shared sex-specific genes between brain and liver. In (F) and (G), four genes related to gamete generation and three Y-linked genes were also shown, respectively. (H) Significant GO terms enriched on the shared male-specific genes. Rich factor represents the proportion of sex-specific genes (male-specific and female-specific genes) or shared sex-specific genes in a GO term to the total genes annotated in the same GO term. Significantly enriched gene ontology (GO) terms were determined with corrected $p$-value using the Banjamini-Hochberg multiple test correction procedure and $q$-value < 0.05. Log ($q$-value) of −1.3 is equal to $q$-value of 0.05.

**Table 2** **Summary of sex-specific and sex-biased genes identified between the sexes in the brain and liver.**

| Tissue | | Brain | Liver |
|---|---|---|---|
| Sex-specific | Male-specific | 133 (1.0%) | 230 (2.0%) |
| | Female-specific | 232 (1.7%) | 458 (4.0%) |
| | Total | 365 (2.7%) | 688 (6.0%) |
| Sex-biased | Male-biased | 1567 (11.6%) | 658 (5.7%) |
| | Female-biased | 1539 (11.4%) | 862 (7.5%) |
| | Total | 3106 (23.0%) | 1520 (13.2%) |
| DEGs | Male | 1700 (12.6%) | 888 (7.7%) |
| | Female | 1771 (13.2%) | 1320 (11.5%) |
| | Total | 3471 (25.8%) | 2208 (19.2%) |

division, meiotic cycle, gamete generation, and humoral immune response (Table S3). In the liver, male-specific genes were enriched into 26 significant GO terms and one KEGG pathway that were mainly involved in regulation of neurotransmitter levels, axon development, and synaptic signaling (Fig. 2D and Table S3). It was notable that these male-specific genes were also enriched into GO terms that were related to digestion and feeding behavior (not significant, but uncorrected $P < 0.01$, Table S3). For female-specific genes, they were enriched into 16 significant GO terms and most were involved in adaptive immune response and regulation of nuclear division (Fig. 2E and Table S3).

To investigate whether different tissues have functional similarities of sex difference, we compared the lists of sex-specific genes identified in the brain and liver. We found 27 male-specific genes and 29 female-specific genes shared by brain and liver (Figs. 2F and 2G, Table S2). Functional enrichment analysis on 27 shared male-specific genes revealed four significant GO terms and all of them were related to digestion (Fig. 2H and Table S4). Interestingly, three of shared male-specific genes (*KDM5D*, *DDX3Y* and *EIF1AY*) are located on the Y chromosome and two of them (*KDM5D* and *DDX3Y*) belong to ancestral Y-linked genes (*Couger et al., 2021*). It was notable that the expression level of *KDM5D* in the brain was over six-fold higher than in the liver, whereas the expression levels of other two Y-linked genes were similar in these two tissues (Table S2). Functional enrichment analysis on the 29 shared female-specific genes did not identify significant GO terms or pathways. However, we found that four of them (*FOXL3*, *GTSF1*, *TMPRSS12*, and *YBX2*) were associated with gamete generation, which was consistent with the enrichment analyses on female-specific genes identified in the brain and liver, respectively (see above).

## Identification and characterization of sex-biased genes

In the brain, a total of 3106 sex-biased genes (SBGs) were identified with similar numbers of male-biased and female-biased genes, whereas in the liver, a total of 1520 SBGs were found with more number of female-biased genes than male-biased genes (Figs. 3A–3D and Table 2). Detailed sex-biased genes have been described in Table S2.

Functional enrichment analysis on female-biased genes in the brain identified 128 significant GO terms and 16 KEGG pathways and most of them were involved in cytoplasmic translation, ATP synthesis coupled oxidative phosphorylation process, ribosome biogenesis, and RNA splicing (Fig. 3E and Table S5). Male-biased genes identified in the brain were enriched into 246 significant GO terms and 19 KEGG pathways and most of them were associated with synaptic signaling, axonogenesis, regulation of cell development and growth, actin cytoskeleton organization, learning and cognition, positive regulation of cellular catabolic process, and circadian regulation of gene expression (Fig. 3F and Table S5). Similar to female-biased genes in the brain, functional enrichment analysis on female-biased genes in the liver revealed 182 significant GO terms and 23 KEGG pathways and most of them were involved in cytoplasmic translation, ATP synthesis coupled oxidative phosphorylation process, and ribosome biogenesis (Fig. 3G and Table S5). In the liver, we found similar functional categories on sex-biased genes as in the brain above. Specifically, male-biased genes in the liver were enriched into 301 significant GO terms and 54 KEGG pathways and they were mostly associated with cellular catabolic

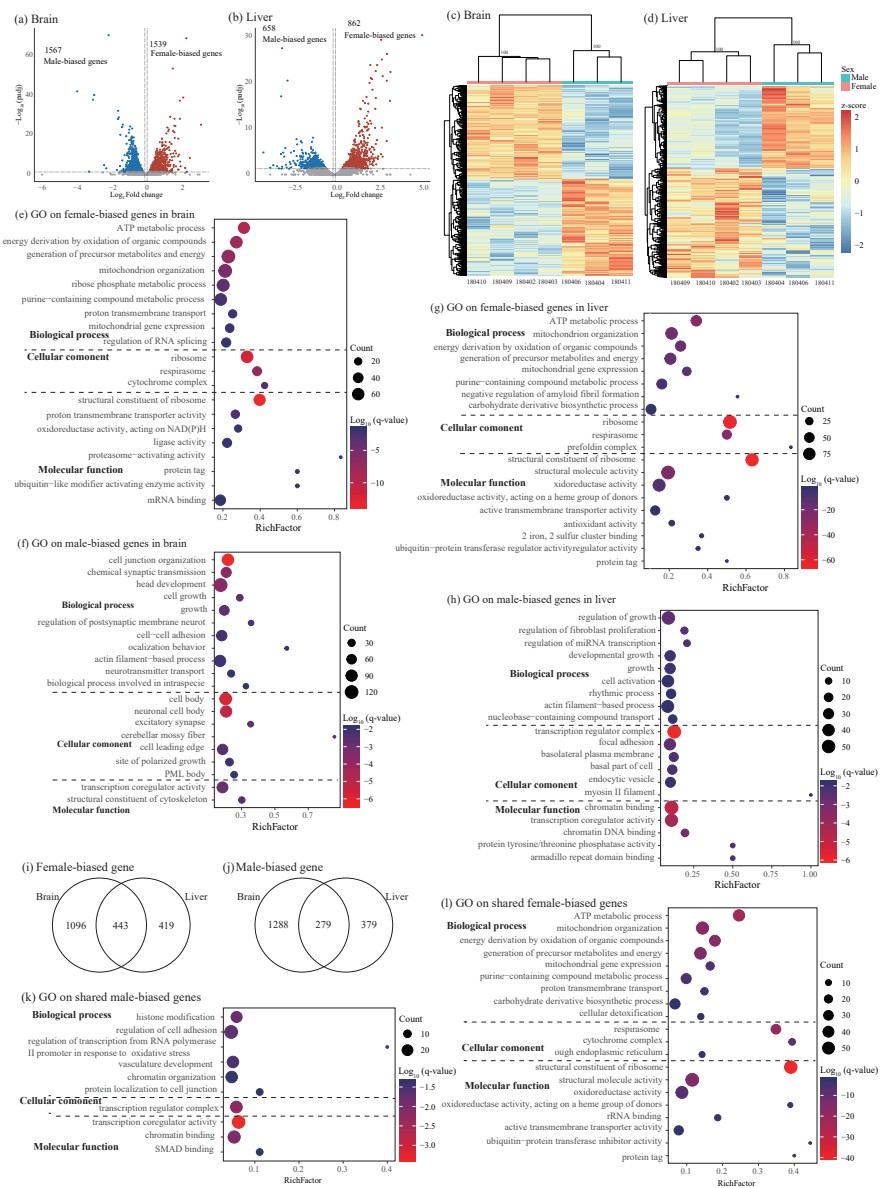

**Figure 3** **Identification and characterization of sex-biased genes.** (A–B) Volcano plots showing sex-biased gene expression in the brain (A) and liver (B). (C–D) Hierarchical clustering and heatmaps showing expression patterns of sex-biased genes in the brain (C) and liver (D). Numbers on each node indicate the bootstrap support values. (E–H) Significant Gene Ontology (GO) terms enriched on sex-biased genes in brain (E, female-biased genes; F, male-biased genes) and liver (G, female-biased genes; H, male-biased genes). (I–J) Venn diagrams showing the number of shared sex-biased genes between brain and liver. (K–L) Significant GO terms enriched on the shared genes (K, male-biased genes; J, female-biased genes). Rich factor represents the proportion of sex-biased genes (male-biased and female-biased genes) or shared sex-biased genes in a GO term to the total genes annotated in the same GO term. Significantly enriched gene ontology (GO) terms were determined with corrected *p*-value using the Banjamini-Hochberg multiple test correction procedure and *q*-value < 0.05. Log (*q*-value) of −1.3 is equal to *q*-value of 0.05.

process, response to hormone and nutrient levels, regulation of growth and fibroblast proliferation, circadian rhythm, and immune function (Fig. 3H and Table S5).

Similar to the analysis on sex-specific genes above, we also compared the lists of sex-biased genes identified in brain and liver and found 722 shared SBGs, including 279 male-biased genes and 443 female-biased genes (Figs. 3I and 3J). Interestingly, we also found 12 SBGs which have opposite expression patterns between the two tissues. Specifically, seven of them were female-biased in the brain but male-biased in the liver; five of them were male-biased in the brain but female-biased in the liver (Table S2). Functional enrichment analysis on 279 shared male-biased genes identified 57 significant GO terms and 7 KEGG pathways and most of them were related to regulation of mRNA catabolic process and stability, hemopoiesis, immune system development, and chromatin organization (Fig. 3K and Table S6). For 443 shared female-biased genes, they were enriched into 144 significant GO terms and 18 KEGG pathways which were mostly associated with energy production *via* oxidative phosphorylation in the mitochondria and ribosome biogenesis (Fig. 3L and Table S6). This was consistent with the separate enrichment analyses on female-biased genes in the brain and liver, respectively (see above).

## Alternative splicing analysis

Using rMATs, we found lots of alternative splicing events between sexes in two somatic tissues. Similar to previous studies (*e.g.*, *Rogers, Palmer & Wright, 2021*), MXE and SE are more common than other three types of splicing in both brain and liver (Table 3). Hierarchical clustering analysis classified males and females into different clusters in both tissues (Figs. 4A and 4B). As for differentially spliced genes (DSGs) between sexes, we found over twice number of DSGs in the brain than in the liver (2231 and 1027 in the brain and liver, respectively, Table 3 and Table S7). Functional enrichment analysis on DSGs in the brain revealed 84 significant GO terms and four KEGG pathways which were mostly related to synaptic signaling, cognition or learning, regulation of RNA splicing and mRNA processing (Fig. 4C and Table S8). In the liver, DSGs were enriched into 180 significant GO terms and 20 KEGG pathways and most of them were involved in catabolic and metabolic processes, regulation of RNA splicing and mRNA processing, humoral immune response, and regulation of coagulation (Fig. 4D and Table S8). By comparing the lists of DSGs in the brain and liver, we found 387 DSGs shared by these two tissues (Fig. 4E) which were enriched into 13 significant GO terms mostly associated with mRNA metabolic process and regulation of RNA splicing (Fig. 4F and Table S9).

## Comparisons of gene differential expression and alternative splicing

To compare the two forms of gene expression regulation, we first explored the difference of chromosomal distribution patterns for DEGs and DSGs. We found that DEGs in females were significantly enriched on the X chromosome in both brain and liver, whereas DEGs in males were less enriched in either brain or liver (Table 4 and Figs. 5A and 5B). For all DEGs, significant enrichment on the X chromosome was observed in the brain but not in the liver. Contrast to the case in DEGs, we did not observe significant enrichment of DSGs on the X chromosome in either brain or liver (Table 4 and Figs. 5A and 5B).

**Table 3** Summary of alternative splicing (AS) events and differentially spliced genes (DSGs) identified between the sexes in the brain and liver.

| Tissue | | Brain | Liver |
|---|---|---|---|
| Splicing events | A3SS | 336 | 189 |
| | A5SS | 341 | 136 |
| | MXE | 1766 | 912 |
| | RI | 391 | 192 |
| | SE | 1113 | 432 |
| | Total | 3940 | 1861 |
| DSGs | A3SS | 273 | 145 |
| | A5SS | 288 | 114 |
| | MXE | 1202 | 548 |
| | RI | 336 | 154 |
| | SE | 787 | 292 |
| | Total | 2231 (16.6%) | 1027 (8.9%) |

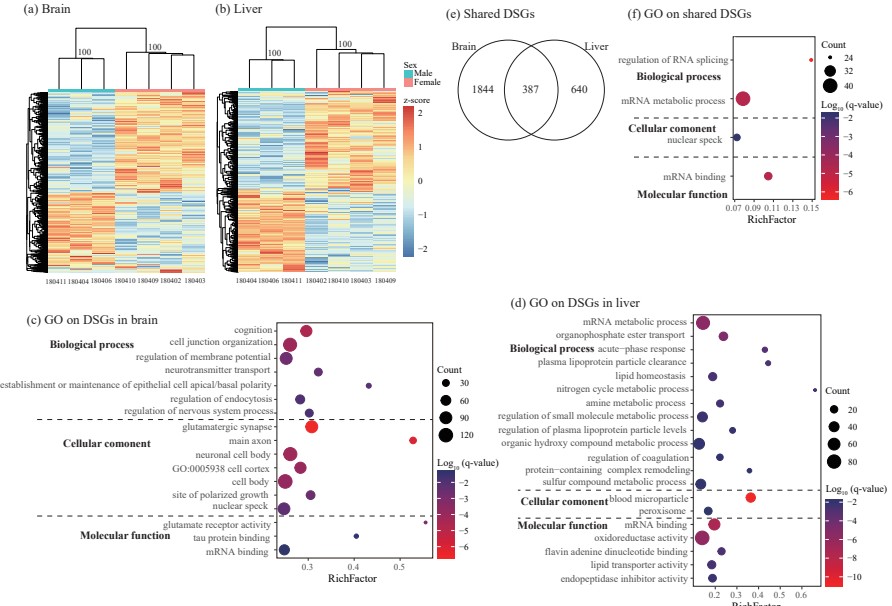

**Figure 4** **Characterization of differentially spliced events and differentially spliced genes (DSGs).** (A–B) Hierarchical clustering and heatmaps showing alternative splicing level in the brain (A) and liver (B). This analysis was based on Euclidean distances of the PSI value of each DSG. The PSI value (percent spliced-in value) represents the proportion of an isoform in one group to the other group at each splice site, ranging from 0 to 1. Numbers on each node indicate the bootstrap support values. (C–D) Significant Gene Ontology (GO) terms enriched on DSGs in brain (C) and liver (D). (E) Venn diagrams showing the number of shared DSGs between brain and liver. (F) Significant GO terms enriched on the shared DSGs. Rich factor represents the proportion of DSGs in a GO term to the total genes annotated in the same GO term. Significantly enriched gene ontology (GO) terms were determined with corrected $p$-value using the Banjamini-Hochberg multiple test correction procedure and $q$-value $< 0.05$. Log ($q$-value) of $-1.3$ is equal to $q$-value of 0.05.

**Table 4   Tests for enrichments of DEGs and DSGs on the X chromosome using Fisher's exact test.**

| Tissue | | | Observed | Expected |
|---|---|---|---|---|
| Brain | DEGs-female | Autosomal | 1675 | 1705.46 |
| | | X-linked | 96 | 65.54 |
| | | *p* value of Fisher's exact test | 0.000 | |
| | DEGs-male | Autosomal | 1643 | 1637.08 |
| | | X-linked | 57 | 62.92 |
| | | *p* value of Fisher's exact test | 0.450 | |
| | DEGs | Autosomal | 3318 | 3342.54 |
| | | X-linked | 153 | 128.46 |
| | | *p* value of Fisher's exact test | 0.012 | |
| | DSGs | Autosomal | 2163 | 2148.43 |
| | | X-linked | 68 | 82.57 |
| | | *p* value of Fisher's exact test | 0.075 | |
| Liver | DEGs-female | Autosomal | 1273 | 1275.36 |
| | | X-linked | 47 | 44.64 |
| | | *p* value of Fisher's exact test | 0.000 | |
| | DEGs-male | Autosomal | 864 | 857.97 |
| | | X-linked | 24 | 30.03 |
| | | *p* value of Fisher's exact test | 0.329 | |
| | DEGs | Autosomal | 2137 | 2133.32 |
| | | X-linked | 71 | 74.68 |
| | | *p* value of Fisher's exact test | 0.694 | |
| | DSGs | Autosomal | 1000 | 992.27 |
| | | X-linked | 27 | 34.73 |
| | | *p* value of Fisher's exact test | 0.175 | |

Notes.
  Abbreviations: DSGs, differentially spliced genes; DEGs, differentially expressed genes, included both sex-specific genes and sex-biased genes; DEGs-female, female-specific genes and female-biased genes; DEGs-male, male-specific genes and male-biased genes.

Second, we test whether there is more overlap than expected between DEGs and DSGs. We observed significant overlap between these two categories of genes in the brain (RF = 1.21, $P < 0.05$) but not in the liver (RF = 0.92, $P > 0.05$, Figs. 5C and 5D). To explore the functional differences between overlapped and non-overlapped DEGs and DSGs in each tissue, we also performed enrichment analyses on each set of genes (Table S10). Specifically, in the brain, we found that the overlapped DEGs and DSGs were mostly involved in the regulation of RNA splicing and synaptic signaling, whereas the only DEGs were in the processes of cytoplasmic translation, oxidative phosphorylation, ATP synthesis, and ribosome biogenesis, and the only DSGs were in synaptic signaling (Table S11). In the liver, we found that overlapped DEGs and DSGs were mostly associated with metabolic and biosynthetic processes, regulation of RNA splicing, cytoplasmic translation, whereas only DEGs were enriched into similar GO terms with only DEGs in brain, and only DSGs were involved in the processes of metabolism and biosynthesis (Table S11).

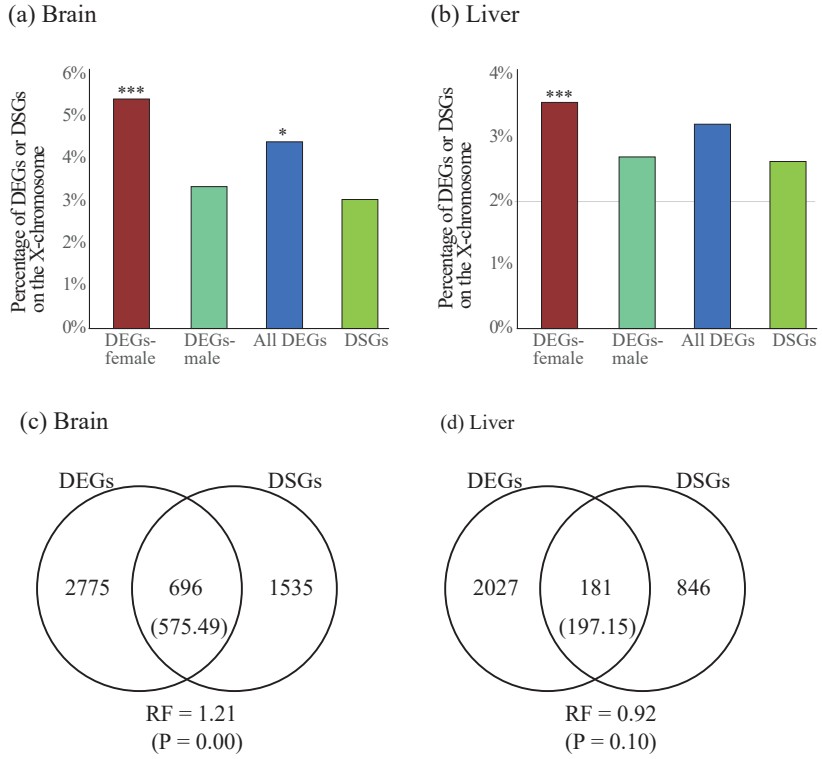

**Figure 5** **(A–B) Enrichment of differentially expressed genes (DEGs) and differentially spliced genes (DSGs) between the sexes on the X chromosome in the brain (A) and liver (B). (C–D) Venn diagrams showing the overlap of DEGs and DSGs in the brain (C) and liver (D).** Numbers in brackets are the expected number of overlapped DEGs and DSGs. DEGs-female: female-specific and female-biased genes; DEGs-male: male-specific and male-biased genes. $*P < 0.05$, $***P < 0.001$.

## DISCUSSION

In this study, we used RNA-seq data of brain and liver, for the first time, to investigate sex differences of gene expression and splicing in bats, a group of mammals exhibiting diverse sexually dimorphic traits (see also in Introduction). Below, we first discussed the results of differential expression analysis and alternative splicing analysis, respectively. Then, we assessed the relative role of these two forms of gene regulation in sex differences.

### Sex differences in differential gene expression

In April, bats arouse from hibernation for feeding and nutrition. Additionally, female bats need to prepare for reproduction, including gamete generation, fertilization and gestation (*Oxberry, 1979*). Consistent with the physiological differences between sexes, we found that female-specific genes in both tissues were mostly involved in nuclear division and gamete generation although the later functional category was not significantly enriched (uncorrected $p < 0.01$). Among them, four (*FOXL3*, *GTSF1*, *TMPRSS12*, and *YBX2*) should be notable here. *FOXL3* is a germ cell-intrinsic factor and it has been shown to be involved in spermatogenesis and the initiation of oogenesis in female gonad of fishes (*Nishimura*

et al., 2015; Kikuchi et al., 2020). GTSF1, encoding gametocyte specific factor 1, has been suggested to play important roles in postnatal oocyte maturation and prespermatogonia in mammals (Krotz et al., 2009; Liperis, 2013; Yoshimura et al., 2018). In mice, TMPRSS12, encoding transmembrane serine protease 12, has been found to be required for male fertility (Zhang et al., 2022) and sperm motility and migration to the oviduct (Larasati et al., 2020). Last, YBX2, encoding Y-box binding protein 2, has been proved to be important in spermatogenesis in mice (He et al., 2019) and also in human (Hammoud et al., 2009). In addition, a majority of female-biased genes in both tissues were associated with cytoplasmic translation and ATP synthesis coupled oxidative phosphorylation process, which provides energy demand for reproduction. Overall, our current study identified thousands of differentially expressed genes between sexes (sex-specific and sex-biased genes) in two somatic tissues which largely contribute to sex differences in physiology (e.g., female reproduction). Thus, our results in bats support the well-known proposal that most sex differences are caused by sex-biased gene expression (Ellegren & Parsch, 2007; Mank, 2017).

Also we found three notable Y-linked genes (KDM5D, DDX3Y and EIF1AY) among the list of male-specific genes in both tissues. KDM5D encodes a histone demethylase for H3K4 demethylation. This gene has also been identified as a male-specific gene and is required for other sexually dimorphic genes in mouse embryonic fibroblasts (Mizukami et al., 2019). A recent study indicated that the X chromosome paralog of KDM5D, KDM5C, could be considered as a determinant of sex difference in adiposity due to its dosage difference between sexes (Link et al., 2020). Here, KDM5C was also identified as a female-biased gene in the brain, suggesting that this gene might also contribute to the sex difference in the brain in bats. DDX3Y (also known as DBY) encodes an ATP-dependent RNA helicase and its main function is related to RNA metabolism. This gene has been shown to be expressed widely across human tissues (Uhlén et al., 2015) and has been suggested to play an important role in dimorphic neural development (Vakilian et al., 2015). These combined results provide further evidences on the contribution of Y chromosome genes beyond sex determination and support their important roles in sexual dimorphic traits of adult nonreproductive tissues (see also Meyfour et al., 2019; Godfrey et al., 2020).

## Sex differences in alternative splicing

Similar to previous studies in other animals (e.g., Drosophila, Gibilisco et al., 2016; birds, Rogers, Palmer & Wright, 2021; human, Trabzuni et al., 2013; Karlebach et al., 2020), we also detected a large number of sex-biased spliced genes in bats (16.6% and 8.9% of expressed genes in the brain and liver, respectively). These combined evidences from different animals and tissues suggest that similar to sex-biased gene expression, sex-biased alternative splicing might be also an important form of gene regulation in encoding sex differences (Karlebach et al., 2020; Singh & Agrawal, 2021).

Although somatic tissues were used in this study, we still observed strong tissue effects on alternative splicing between sexes with over twice number of DSGs identified in the brain than in the liver. This tissue effects of sex-biased splicing has also been reported in previous studies in birds (Rogers, Palmer & Wright, 2021) and Drosophila (Gibilisco et al., 2016). However, in both previous studies, gonad and somatic tissues were used and they

found little sex-biased splicing in somatic tissues comparing to gonad tissues (*Gibilisco et al., 2016*; *Rogers, Palmer & Wright, 2021*). Further evidences of tissue differences between somatic and gonad tissues was from the hierarchical clustering analysis based on alternative splicing level in *Rogers, Palmer & Wright (2021)*, where males and females were mixed in the somatic tissue but they clustered separately in the gonad tissues. However, our hierarchical clustering analysis revealed that both somatic tissues showed clustering between males and females. The difference between these two studies might be resulted from tissue effect on different somatic tissues. Indeed, a recent study on 39 different tissues in human revealed that a majority of alternative splicing events (97.6%) were specific to one tissue (*Karlebach et al., 2020*).

## Comparisons of the two forms of gene expression regulation

Our results showed that in both somatic tissues (brain and liver), DEGs in females (female-specific and female-biased genes) were found to be more enriched than expected in X chromosome, which is similar to previous studies in other organisms (*e.g.*, fish, *Leder et al., 2010*; *Sharma et al., 2014*; water strider, *Toubiana et al., 2021*; mouse, *Khil et al., 2004*; *Yang et al., 2006*; human, *Oliva et al., 2020*). Enrichment of sex-biased genes in X chromosome has been proposed to resolve sexual conflict or sexual dimorphism (*Rice, 1984*; *Rice, 1987*; *Rowe, Chenoweth & Agrawal, 2018*) although this proposal has been recently questioned (*Ruzicka & Connallon, 2020*).

Contrast to the case of DEGs, we did not observe a significant enrichment of DSGs in X chromosome. Up to now, less studies have been performed to investigate the genomic distributions of sex-biased DSGs. In addition, those few published studies revealed different results. A recent study based on combined results of 39 tissues found that sex-biased DSGs were significantly enriched in X chromosome (*Karlebach et al., 2020*). However, another recent study on different tissues of *Drosophila* found that sex-biased DSGs identified in the whole body were enriched in X chromosome while ones in the head were not enriched (*Singh & Agrawal, 2021*). We proposed that the inconsistency between different studies might be largely caused by different tissues used because there was a strong tissue effect on sex-biased alternative splicing (*Karlebach et al., 2020*).

We observed more than expected overlap of DEGs and DSGs identified between the sexes in the brain but less than expected overlap in the liver. This contrast result might be caused by the difference of the extent of complexity between the two tissues. Compared to liver, the brain is more complex and more involved in sex differences. Indeed, we observed more number of DEGs and DSGs in the brain than the liver (brain: 3471 DEGs and 2231 DSGs; liver: 2208 DEGs and 1027 DSGs). Again, the previous studies on the extent of overlap between the two sets of genes revealed different results. In *Rogers, Palmer & Wright (2021)*, less than expected overlap of DEGs and DSGs was observed in the gonad. However, in *Karlebach et al. (2020)*, the authors observed more than expected overlap between these two sets of genes. This inconsistency between different studies might also result from tissue specificity in sex-biased gene expression or alternative splicing possibly due to the difference of the extent of complexity across tissues.

Overall, our current results, combined previous studies, suggested that the relative roles of differential gene expression and alternative splicing in sex differences may have tissue specificity. In addition, we found that the only DEGs and only DSGs in each tissue were enriched into different functional categories. Thus, our study further supports that the two forms of gene regulation might play complementary roles in encoding sex differences (*Rogers, Palmer & Wright, 2021*; *Singh & Agrawal, 2021*; *Karlebach et al., 2020*).

### Limitations of the study

In this study, we identified far more DSGs between males and females than DEGs in both brain and liver, whereas a recent study detected far fewer DSGs between sexes than DEGs in birds (*Rogers, Palmer & Wright, 2021*). This contrast may be resulted from different kinds of tissues used between studies (reproductive tissue in (*Rogers, Palmer & Wright, 2021*) while somatic tissues in this study). In the future reproductive tissues of our study system will be used to test whether there were different effects of differential expression and splicing on sex-related regulatory networks between reproductive and nonreproductive tissues.

To make comparable analysis on gene expression patterns, individuals of this study were collected in the same population and at the same time. However, we still cannot confidently determine whether the sampled individuals were at the same age. To reduce the effect of age on gene expression, we only used adult bats in this study (*Chen & Mao, 2022*). In the future, we can first determine the age of bats using DNA methylation profiles which use noninvasive sampling (*Wilkinson et al., 2021*). Then, bats with the same age were used to assess the sex differences of gene expression and splicing.

Similar to the majority of current studies on gene expression and splicing, here we used bulk RNA-seq which may mask difference of gene expression and splicing between the sexes because this sequencing strategy assess the difference of expression using the average level of multiple cell types in the tissue. In the future, single-cell transcriptome analyses (*Kulkarni et al., 2019*) will be promising to explore the difference of sex-biased gene expression and splicing in different cell types (*Kasimatis, Sánchez-Ramírez & Stevenson, 2021*). In addition, it will be interesting to examine specific regions of the brain to determine differentially expressed and spliced genes in males and females in the future. Finally, it is difficult to reconstruct isoforms with short-read RNA-seq. In the future, we can identify sex-specific transcripts accurately using long-read RNA-seq (*e.g.*, PacBio Iso-Seq) which can skip the bioinformatics steps of reconstructing isoforms (*e.g.*, in fishes, *Naftaly, Pau & White, 2021*).

## CONCLUSIONS

In two somatic tissues of bats, we found many differentially expressed genes between the sexes which largely contributed to their physiological differences. In addition, our results in bats also support an important role of sex-biased alternative splicing in sex differences. As for the relative roles of these two gene regulation forms, it may depend on specific tissues used in the study.

## ACKNOWLEDGEMENTS

We thank Sun Haijian, Wang JY, and Ding YT for assistance with sample collection. We also thank Kush Shrivastava, Fernando Diaz and four anonymous reviewers for constructive comments that improved the manuscript.

### Funding

This work was supported by The Scientific and Technological Innovation Plan of Shanghai Science and Technology Committee (20ZR1417000). The funders had no role in study design, data collection and analysis, decision to publish, or preparation of the manuscript.

### Grant Disclosures

The following grant information was disclosed by the authors:
The Scientific and Technological Innovation Plan of Shanghai Science and Technology Committee: 20ZR1417000.

### Competing Interests

The authors declare there are no competing interests.

### Author Contributions

- Wenli Chen conceived and designed the experiments, performed the experiments, analyzed the data, prepared figures and/or tables, authored or reviewed drafts of the article, and approved the final draft.
- Weiwei Zhou and Qianqian Li analyzed the data, prepared figures and/or tables, and approved the final draft.
- Xiuguang Mao conceived and designed the experiments, authored or reviewed drafts of the article, and approved the final draft.

### Animal Ethics

The following information was supplied relating to ethical approvals (*i.e.*, approving body and any reference numbers):

National Animal Research Authority, East China Normal University

### Data Availability

All data are available in the Supplemental Files and the sequencing reads are available at NCBI Sequence Read Archive (SRA): PRJNA763734.

### Supplemental Information

Supplemental information for this article can be found online at http://dx.doi.org/10.7717/peerj.15231#supplemental-information.

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
