# Peer review of "Sex differences in gene expression and alternative splicing in the Chinese horseshoe bat"

_PeerJ, doi:10.7717/peerj.15231_

## Round 0.1 · original submission · Minor Revisions

Dear Authors,

Thank you for submitting your manuscript to PeerJ: Life & Environment. I am having full review reports about your manuscript. It seems that in general the views of the reviewers are positive. However, before the final acceptance of the article I wish that you should make some necessary changes as per reviewers suggestion. I suggest, please go through the detailed comments from Reviewers 2,3 and 4 and prepare your revised manuscript accordingly. Some of my concerns are enlisted by Reviewers 2 and 3, which I hope you will incorporate in your revised manuscript. As also pointed out by reviewer 2, why the KEGG analysis was not performed? It is necessary that results already pointed out in table should not be re-written in text, they should only be discussed.

I would also like to know the rationale behind choosing brain and liver tissues.

As pointed out by reviewer 3 also I am not able to open figure S1.

As minor note -

1. Although the language part is ok, please check for minor grammatical corrections.
2. Please address comments from reviewers 2, 3 & 4. As reviewer 1 has given no comments kindly don't include it in your revised manuscript.

Thanks and Regards,

Reviewer 1 ·

Basic reporting

Clear and unambiguous

Experimental design

Research questions well defined

Validity of the findings

Conclusions are well stated

Reviewer 2 ·

Basic reporting

This is an interesting study about the sex differences in gene expression and splicing in Rhinolophus sinicus. The samples are collected from wild caught bats in April when the bats just wake from hibernation. The authors compared the brain and liver transcriptomes of males and females. Their results support that differential gene expression and alternative splicing are both important and they may play complementary roles in encoding sex differences. While I still have few comments. Please see additional comments.

Experimental design

In the methods, the authors just performed the GO analyses. Why do they perform the KEGG pathway enrichment?

Validity of the findings

no comment

Additional comments

1. In the first and second paragrahs of the introduction section, the authors described differential gene expression and alternative splicing are used to analyzed sexual differences of animals. While I suggest the authors could summary the main results of those studies and describe the limitations of those studies.
2. In L56, the authors referred that reproductive and somatic tissues were used to studied. Please indicate the advantages and disadvantages between them because this study is based on somatic tissues.
3. In L188-193, the contents are not results. Some of them need to be described in methods section.
4. In L228-236, these belong to the discussion. It should not be in the results section.
5. In L242-250, these are still discussion.
6. In the limitations section, I suggest the authors discuss the limitation of tissue selection.
7. Some references need to be corrected to the same form based on the requirement of the journal.

·

Basic reporting

All criteria are met here!

This study assesses the role of gene expression vs. alternative splicing mechanisms to transcriptional change between sexes using brain and liver samples in bats. For this, the authors use RNA-Seq data from a previous study (4 individuals per sex and tissue). While the previous study investigates gene expression changes associated with seasonality, the current study introduces alternative splicing analyses to investigate sex differences using the same data set. The main finding, as it is common in these studies, is that thousands of genes show signatures of differential gene expression (DEGs), with alternatively spliced genes (DSGs) showing as much variation as differentially expressed genes. DEGs were significantly associated with the X chromosome, while DSGs were not. Finally, DEGs and DSGs showed significant overlap in the brain but not in the liver.

This study is well-written, easy to read, and follows standard and robust bioinformatic analyses in the field of transcriptomics. It also contributes significantly to understanding the extent of alternative splicing mechanisms in defining sex differences.

Experimental design

All criteria are met here!

Validity of the findings

Although the general approach and most of the author's claims are well supported with available evidence, I have raised a number of concerns where I think more clarity is needed. I also found some potential bias in the authors' interpretations of functional analyses. Here, I explain these concerns and make some suggestions to help improve the manuscript for its final publication.

1) The reference genome:
RNA-Seq reads were mapped to the R. sinicus reference genome (line 111). However, the citation for this genome is a personal communication "(Ji Dong lab, personal communication)", also referenced as unpublished data in the previous paper (Chen & Mao 2022). I am not sure why the appropriate citation or repository for the genome is not provided. Is the genome of this species the same as that of Dong et al. (2016)? Has it been modified over the years? I strongly recommend explaining the state and quality of the genome used as a reference, including the corresponding citations and further modifications, if any. The X chromosome analysis suggests that the genome is good quality and even at the chromosome level. If it has been modified, please, introduce a brief description that helps the reader understand the structure and annotation level of the reference.

2) Basic statistics:
As mentioned above, the methods and results sections, particularly the bioinformatic analyses, follow the standard and robust practices in the field of transcriptomics. The basic Illumina sequencing results are included in Table S1 but are not mentioned in the results. The authors should indicate at least the average raw-read number and the number of remaining reads after the whole filtering process to give the reader a good idea about the level of resolution of the whole study. The mapping rates are also crucial as an additional indication of the genome quality. These could be minor edits to the text, as the data is already in the supplementary tables. Also, it is not clear what "Clean" means in Table S1.

In addition, a power analysis is mentioned in methods (lines 114-117), which seems like a nice approach regarding the previous concern, but I failed to find these results in the manuscript. Also, in the results, 13456 and 11502 genes were expressed in brains and livers, respectively (lines 196-200). Please, indicate how many genes are annotated in the reference genome.

3) Interpretation of functional analyses and reproductive-related genes
Several GO categories were associated with the different sets of DEGs and DSGs genes in brain and liver tissues. The results of these analyses are very well explained in the Results section; however, I found several biased interpretations throughout the manuscript. The clearest example is the exaggerated role of reproductive-related categories in defining sex differences, which are mentioned in the Abstract, the final part of the Introduction, and the Discussion sections. Although these categories are more interesting in terms of the fundamental question of the paper (transcriptional bases of sex differences), these are also the most underrepresented genes according to the functional results. It seems like most of these genes are not even significant after FDR corrections, and some others are based on the identification of individual genes. Ignoring the FDR corrections to highlight the role of reproductive genes when no enrichment is found (lines 207, 212, 216, and 329) makes the reader wonder why the role of the most significant categories is diminished. I suggest removing these comments from the corresponding sections and focusing the interpretations on the most significant categories, which are also interesting, even if they are unrelated to reproduction.

The previous concern might also be associated with the target tissues analyzed in the study. I think the authors did an excellent job making clear that the brain and liver are somatic tissues and discussed some comparisons with other species using reproductive tissues. I do not have the expertise to speak about reproductive physiology in mammals. However, the authors should explain why they expect to find reproductive-related genes driving sex differences in somatic tissues such as the brain and liver. For example, in the final part of the Introduction, it says that they expect to find genes associated with "gamete generation and gestation". This rationale or explanation will apply to all the arguments about reproductive genes throughout the manuscript.

In my opinion, some of the sex differences in expression could be indirectly associated with reproduction. For example, if some of the most significant genes are associated with digestion or energy demand, as discussed in lines 329-332, and there is evidence for higher energy investment in females than in males, the speculation would be better grounded on the statistical evidence.

Additional comments

It is indicated in the methods that "Log (q-value) of -1.3 is equal to q-value of 0.05" (line 166). Please, include some information about the FDR corrections and the meaning of the Log10 used in the figures. It will help the reader understand the figures without going back to the methods section.

I think the figures look great and are very well organized, following the analyses and story. However, some of the panels are very large, making the font too small. It is challenging to read some of the figures without zooming in multiple times. This is more of an editorial concern, I think.

Line 370: remove "two".

I could not open Figure S1.

Reviewer 4 ·

Basic reporting

The language used is easy to understand, with very few grammatical mistakes. References cited support the background and findings of the present study.

Experimental design

The experiment is well designed, with all the necessary experiments performed and tools used. The lacunae of the study are also pointed out along with the scope for future studies.
Suggestion: It is known that different parts of the brain are responsible for different functions of the body. Will it be possible to target specific regions of the brain to identify DEGs in males and females?

Validity of the findings

The raw data provided supports the findings of the study.

Additional comments

As the lacunae have been pointed out by the authors, age of the animals needs to be identified before conducting the main experiments to determine DEGs and DSGs. Determination of age by DNA methylation studies would require more number of bats to be euthanized and their DNA screened. Will it be ethical to slaughter these many wild animals for experimentation? If possible other method/ way to determine the age of the animal should be followed.

Annotated reviews are not available for download in order to protect the identity of reviewers who chose to remain anonymous.

---

## Round 0.2 · Minor Revisions

Dear Authors,

Thank you for submitting your revised manuscript to PeerJ. I am having the full review reports of reviewers for your article, which are generally positive. The revised article is now more readable and interesting. The overall construct of the paper. data quality and analysis part is now coming out nicely. However, I would like to request you put a little more effort to make it more understandable & explanatory so that readers from a broader scientific field can enjoy it.

The complete reviewers' reports and their suggestions are attached to this email. Please revise your manuscript and prepare the response accordingly.
Reviewers 2 & 3 have suggested minor modifications to include in the revised draft. Should you choose not to include these suggestions in the current draft, please state clear reason in your response letter.

Suggestions of reviewer 5 are in common with my questions. The importance of species and tissue selection is still missing. There are some accurate corrections also cited by reviewer 5. I would recommend you go through the comments in detail and prepare your response/ revised manuscript accordingly.

Looking forward to your response,

Thanks and Regards,

Reviewer 2 ·

Basic reporting

Most of comments were referred and revised by authors. For the KEGG enrichment analysis, the authors explained why they did not do it. While if possible, I still suggest the authors can perform it. It is possible they can got new results which differed from GO enrichment analysis.

Experimental design

no comment

Validity of the findings

no comment

·

Basic reporting

No comments.

Experimental design

No comments.

Validity of the findings

I only have one final comment regarding the power analysis. This is an excellent point of the study, but the reported result (“RNASeqpower was 0.89”) seems a little incomplete. The power depends on several assumptions, such as coverage, statistical significance, and fold-change thresholds. I recommend including a supplementary figure that summarizes these results instead of just mentioning that the power was 0.89.

Additional comments

I have read the revised manuscript and am happy with how the authors have addressed my comments and concerns. I can now recommend the manuscript for publication.

Reviewer 4 ·

Basic reporting

All the corrections and suggestions, as mentioned by the reviewers have been incorporated by the authors.
The article fulfills all criteria.

Experimental design

Experiments are well designed and enough data is generated

Validity of the findings

The findings are presented in appropriately.

Reviewer 5 ·

Basic reporting

No comment.

Experimental design

No comment.

Validity of the findings

No comment.

Additional comments

Chen et al PeerJ

Chen et al present a nice study on the role of differential expression and AS in bats. While much is known about the role of gene expression in adaptive evolution, less is known about AS, particularly in the realm of sex-specific differences. So, this study will make a valuable addition to this literature.

Major comments:

I feel like throughout the manuscript, the authors failed to convince to why this bat is an important system for studying sexual dimorphism. This is crucial. Please work on this especially in the abstract, intro, and discussion.

The abstract needs some re-working. For example, in line 37 why is it important to study this in this species of bats? What’s special about them? Add a sentence about this. You essentially want to illustrate that this bat is an interesting system for studying sexual dimorphism.

Line74, what are the sexual dimorphisms in this species of bats? Maybe as a photo of male vs female bats and refer to it. I feel like this whole paragraph (L73 – 77) needs to be expanded to cover the biology of these bats, and why they are important models for study sex differences. After reading the manuscript I still don’t know what are the sexually dimorphism is in these bats. What are these physiological differences that you speak of in L323?!

L297 Okay but why did you study these bats?


Figure 1 jumps straight into the results. It would be nice to see an introductory figure showing the photos of male vs female bats, and their sampling location(s), and a PCA/heatmap of all genes across all samples from both tissues analysed in this study. It gives a nice introduction to the system as well as the overall landscape of the transcriptomes.

L399 sexual conflict in what? This is a cool angle and could be developed further in the Discussion.

Can you expand in the Discussion on why “more than expected overlap of DEGs and DSGs was observed in the brain but not in the liver”. Do you think it is because the brain is more complex and more involved in sex-specific differences so each gene is regulated in different ways? What is the function of the overlapping DEG and DSG overlapping genes versus those that are non-overlapping?

Overall, I feel like the Discussion needs more thought and insight. It currently reads like a modified recap of the results. Maybe you could think about the following points:
1. How do your results shed light on how gene regulation resolves sex conflict?
2. Why is it that some genes are regulated by both differential expression and splicing while others aren’t? Why would this be organ specific?
3. It isn’t unexpected that Y-linked genes would be differentially regulated – see sexual conflict theory of Y chromosome evolution
4. What have you learnt overall about sex-specific differences in this bat from your analyses?
5. What have you learnt overall about AS and DGE in bats and how does this related to this beyond bats and sex?
6. What's the big picture here?


Minor comments:

Line 34: Not completely true, many still question the how commonly AS contributes to phenotypic diversity. It certainly does contribute to transcriptional diversity.


L62 Cite and read recent paper on the importance of AS in adaptive evolution (Singh & Ahi 2022) that covers sex-specific differences too


L81 please justify why you picked brain, lived and not gonads or some other relevant organ. It is not enough to say that “because other studies used it”. What happens in the brain or liver that determines sex differences? Maybe set the stage for the interesting GO terms that you find in your results such as cognition or growth etc.

L4227 it is difficult to reconstruct isoforms with short-read data. Pacbio isoseq sequences full isoforms so you can skip the bioinfomatic step of reconstructing isoforms = higher accuracy.

L340 “..beyond sex determination” in sex associated traits such as xxx

Table 3 only has percentages calculated for DSGs, add it for DEGs too.

---

## Round 0.3 · Minor Revisions

Dear Authors,
Thank you for submitting your revised manuscript. I have gone through your revisions and the paper is now readable and more interesting. Our reviewers' reports are also positive for your revised version. The part limitations of the study were interesting to note and well-written by the authors.

However, reviewer 3 has some comments on the presentation of your power analysis and reviewer 5 has suggested a last correction in the form of title and abstract. hence, I am recommending a minor revision, please read the comments of reviewers 3 and 5, should you choose not to incorporate these changes please present a proper rebuttal in the letter.

I am also attaching an annotated manuscript, some grammatical and sentence corrections have been made by me. It is requested that all the authors should again re-check the language & relevant parts before the final submission.

Looking forward to your revised manuscript,
Best Regards,

KS

Reviewer 2 ·

Basic reporting

The authors added the KEGG analysis and got the similar results with GO analysis. They have added the corresponding results in Supporting materials. Now I have no comments with the manuscript.

Experimental design

no comment

Validity of the findings

no comment

·

Basic reporting

All my concerns were revised by the authors.

Experimental design

All my concerns were revised by the authors.

Validity of the findings

All my concerns were revised by the authors.

Additional comments

The power analysis is now included but has no interpretation. The issue now is that the power numbers are much lower than the canonical 80%, (68% for brain and 40% for liver samples). I am not sure how the authors are manipulating the simulation, but my understanding from Table S1 is that the analysis was done with some default parameters (e.g., 100 reads), which is meaningless when compared with all the data on DEG and ASG genes. With the power analysis, you want to show that you have enough statistical power to support your main results above certain thresholds. If you just leave the power analysis with these default parameters, you are basically saying that you don’t have enough power to detect the differences in the results section. Finding those thresholds (of coverage and fold change) and comparing them back to the data is the whole point of this analysis.

That being said, I think this is a minor issue. Based on the coverage reported in Table S2, all samples have > 30 million reads, which tells me that the paper should not have a power issue. So, if the power analysis is not fully interpreted, I suggest removing this form from the paper instead of leaving it with the default parameters. The original issue was that it was mentioned but not shown.

I feel like the authors (and the AE) can easily choose the best way to go and submit their final version without more revisions from my side.

Reviewer 5 ·

Basic reporting

I really appreciate how the authors have incorporated my suggestions. The manuscript has shaped up nicely! Figure 1 is very useful. I just think the map in Figure 1a could be improved in quality.

I have edited the title and the abstract of this manuscript (see below). I would recommend that the authors use this version for their final manuscript.
* * *
Sex differences in gene expression and alternative splicing in the Chinese horseshoe bat

Sexually dimorphic traits are common in sexually reproducing organisms and can be encoded by differential gene regulation between males and females. Although alternative splicing is common mechanism in generating transcriptional diversity, its role in generating sex differences relative to differential gene expression is less clear. Here, we investigate the relative roles of differential gene expression and alternative splicing between male and female the horseshoe bat species, Rhinolophus sinicus. Horseshoe bats are an excellent model to study acoustic differences between sexes. Using RNA-seq analyses of two somatic tissues (brain and liver) from males and females of the same population, we identified 3,471 and 2,208 differentially expressed genes between the sexes (DEGs) in the brain and liver, respectively. DEGs were enriched with functional categories associated with physiological
difference of the sexes (e.g. gamete generation and energy production for reproduction in
females). In addition, we also detected many differentially spliced genes between the sexes (DSGs, 2,231 and 1,027 in the brain and liver, respectively) which were mainly involved in regulation of RNA splicing and mRNA metabolic process. Interestingly, we found a significant enrichment of DEGs on the X chromosome, but not for DSGs. As for the
extent of overlap between the two sets of genes, more than expected overlap of DEGs and
DSGs was observed in the brain but not in the liver. This suggests that more complex tissues, such as the brain, may require the intricate and simultaneous interplay of both differential gene expression and splicing of genes to govern sex-specific functions. Overall, our results support that variation in gene expression and alternative splicing are important and complementary mechanisms governing sex differences.

Experimental design

NA

Validity of the findings

NA

Additional comments

NA

---

## Round 0.4 · accepted · Accept

Dear Authors,

Thank you for submitting your revised manuscript to PeerJ. The revised version is now having all the necessary corrections and is more readable as well as interesting. As per the recommendations of our reviewers in the last revision the manuscript requires no further revision/suggestions to be incorporated and hence it can be accepted for publication. Thank you for your submission to PeerJ, you will be contacted by the publication house for further processing of your accepted manuscript in due time.

With Best Regards,
Kush Shrivastava, PhD